# Time-Series Anomaly Detection Based on Dynamic Temporal Graph Convolutional Network for Epilepsy Diagnosis

**Guanlin Wu** [ID]**, Ke Yu ***, **Hao Zhou, Xiaofei Wu and Sixi Su**

School of Artificial Intelligence, Beijing University of Posts and Telecommunications , Beijing 100876, China; wuguanlin@bupt.edu.cn (G.W.); zhouh@bupt.edu.cn (H.Z.); wuxf@bupt.edu.cn (X.W.); sxsu@bupt.edu.cn (S.S.)
* Correspondence: yuke@bupt.edu.cn

**Abstract:** Electroencephalography (EEG) is typical time-series data. Designing an automatic detection model for EEG is of great significance for disease diagnosis. For example, EEG stands as one of the most potent diagnostic tools for epilepsy detection. A myriad of studies have employed EEG to detect and classify epilepsy, yet these investigations harbor certain limitations. Firstly, most existing research concentrates on the labels of sliced EEG signals, neglecting epilepsy labels associated with each time step in the original EEG signal—what we term fine-grained labels. Secondly, a majority of these studies utilize static graphs to depict EEG's spatial characteristics, thereby disregarding the dynamic interplay among EEG channels. Consequently, the efficient nature of EEG structures may not be captured. In response to these challenges, we propose a novel seizure detection and classification framework—the dynamic temporal graph convolutional network (DTGCN). This method is specifically designed to model the interdependencies in temporal and spatial dimensions within EEG signals. The proposed DTGCN model includes a unique seizure attention layer conceived to capture the distribution and diffusion patterns of epilepsy. Additionally, the model incorporates a graph structure learning layer to represent the dynamically evolving graph structure inherent in the data. We rigorously evaluated the proposed DTGCN model using a substantial publicly available dataset, TUSZ, consisting of 5499 EEGs. The subsequent experimental results convincingly demonstrated that the DTGCN model outperformed the existing state-of-the-art methods in terms of efficiency and accuracy for both seizure detection and classification tasks.

**Keywords:** time-series anomaly detection; graph convolutional network; electroencephalography; seizure detection and classification

## 1. Introduction

Anomaly detection is a well-established research topic that has attracted a lot of research attention for decades. Epilepsy, often seen in children and the elderly [1], manifests as temporary losses of consciousness, convulsions, spasms, or abnormal behaviors, severely disrupting the patient's routine life. Electroencephalography(EEG), a non-invasive technique for diagnosing epilepsy, captures brain neural activities by recording electrical activities on the scalp [2], which is helpful for doctors to diagnose epilepsy, appraise treatment effectiveness, and predict relapses. However, the analysis of EEG signals is a complex and labor-intensive task that requires interpretation and classification by experienced and specially trained medical professionals or technicians. This process consumes a significant amount of time and resources. Accordingly, the development of automatic and computer-assisted algorithms for EEG-based epilepsy detection and classification can accelerate the diagnosis process and curtail treatment costs, bearing substantial practical significance.

The epilepsy seizure process can be categorized into four distinct phases [3]: the interictal, preictal, ictal, and postictal periods. During the interictal period, patients typically exhibit no symptoms. However, as they transition into the preictal stage, prodromal symptoms such as headaches and concentration difficulties may emerge. This phase

potentially progresses to the vital ictal stage, characterized by symptoms, like cramps and convulsions. Finally, in the postictal stage, patients often experience symptoms, including fatigue and headaches. As demonstrated in Figure 1 below, the EEG frequencies markedly differ across these stages.

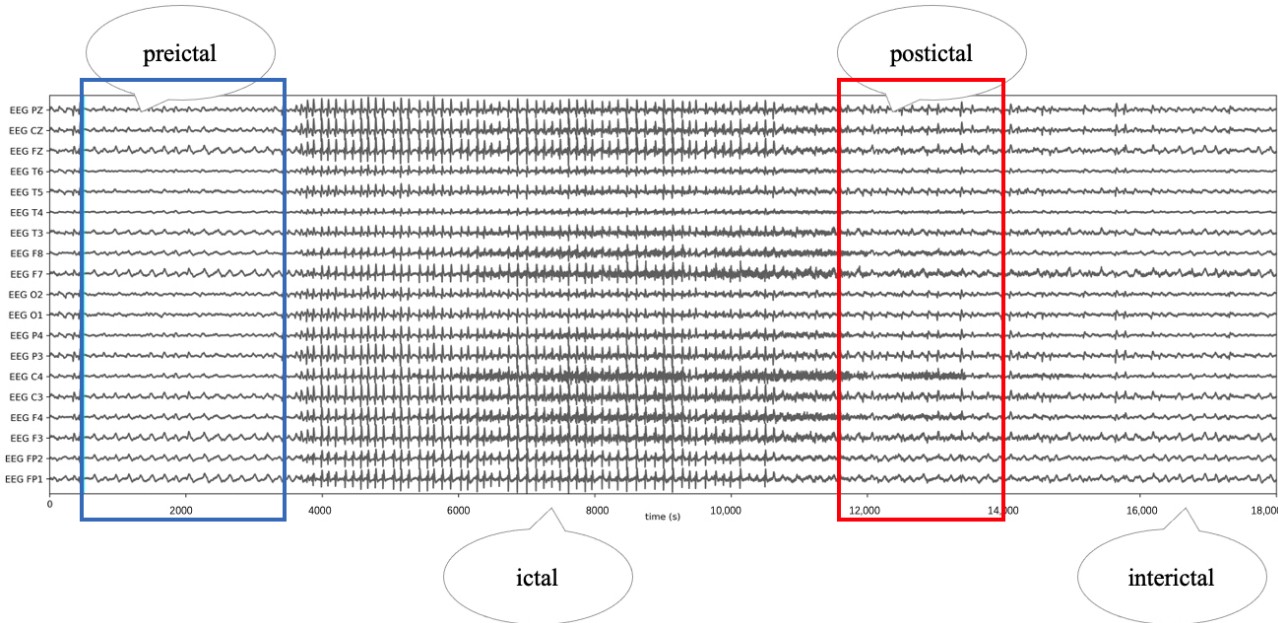

**Figure 1.** Changes in EEG signals at different periods of seizure. The EEG signals in the blue box indicate the preictal period, the EEG signals in the red box indicate the ictal period, the EEG signals in between the two boxes indicate the postictal period, and the EEG signals elsewhere indicate the interictal period.

In the process of EEG data collection, traditional methods typically involve segmenting the EEG signal prior and directly inputting it into the model. The majority of previous studies have employed a standard time interval of either 60 s or 12 s for slicing the signals [4,5]. Subsequent to this slicing process, the segmented signals are labeled as either 'seizure' or 'non-seizure', which is a process termed as the generation of coarse-grained labels. The TUSZ dataset [6,7], on the other hand, provides more detailed labeling by marking the start and end points of epileptic seizures, thus producing labels for each individual time step of the EEG signal [6]. This comprehensive labeling technique is referred to as fine-grained labeling. Nevertheless, the current methodologies primarily focus on the coarse-grained labels generated from segmented EEG fragments [4,5], thereby neglecting the rich information contained in the fine-grained labels of the original, unsegmented EEG signals.

With the advent of deep learning, the current methodologies frequently employ EEG signals as training data, leveraging advanced models for epilepsy-related research. Representative examples include recent studies that utilize convolutional neural networks (CNNs) [8,9] or recurrent neural networks (RNNs) [10,11] for seizure detection and classification tasks. However, these techniques have their limitations. The EEG signals are collected from multiple channels, so the signals are not independent of each other. These methods primarily model the temporal features but tend to ignore their spatial characteristics, leading to a decrease in detection accuracy. In response to this challenge, graph neural networks (GNNs) are introduced to capture the spatial information of EEG data. For instance, Tang et al. [4] proposed two novel EEG graph structures and intended to capture both electrode geometry and brain connectivity. Raeisi et al. [12] integrate temporal features extracted from individual EEG signals with short- and long-range spatial interdependencies among EEG channels. These interdependencies are embedded within a graphical representation of multi-channel EEG data. Meanwhile, Tao et al. [5] incorpo-

rated dynamic networks to encapsulate the spatiotemporal features associated with the connectivity between different brain regions. Despite these innovative approaches yielding promising results, performance is still hindered by the inadequate modeling of dynamic graph structures and insufficient attention to fine-grained labels.

To overcome the aforementioned challenges, we introduce a novel framework, the dynamic temporal graph convolutional network (DTGCN), for the detection and classification of seizure zones. More specifically, we initially employ a seizure attention module coupled with fine-grained labels to model both the distribution and diffusion patterns of epilepsy. Subsequently, we propose a graph structure learning methodology designed to extract dynamically evolving graph structures inherent in multivariate EEG signals. This approach is capable of profiling the intrinsic characteristics of dynamic connections within brain networks. Lastly, we utilize the temporal graph convolution network (TGCN) [13] to extract spatiotemporal features. In conclusion, the key contributions of our study can be summarized as follows:

- We propose a dynamic temporal graph convolutional network (DTGCN) model, which is capable of detecting and classifying epilepsy using fine-grained labels and dynamic graphs. The effectiveness and superiority of our approach are substantiated through ablation studies and visualization experiments conducted on the TUSZ dataset.
- We design a seizure attention module that utilizes fine-grained labels to model the distribution and diffusion patterns of epilepsy and incorporates attention scores into the final loss function. This innovative approach encourages the model to concentrate more efficiently on abnormal time steps.
- We devise a strategy for dynamically generating EEG graph structures using predefined graphs, thereby modeling the dynamic connectivity characteristics within brain networks. Furthermore, the rate of change in the graph structure can be modulated via parameters, enabling more flexible adaptation to varying scenarios.

The structure of the remaining sections of this paper is as follows: Section 2 provides an overview of the related work. In Section 3, we introduce the DTGCN framework in detail. The experimental results and corresponding analyses are presented in Section 4. Finally, we draw conclusions in Section 5.

## 2. Related Work

Over the past several decades, a wealth of research has been devoted to the automated detection and classification of epilepsy. The majority of these studies initially extract statistical features manually, followed by the application of traditional machine learning algorithms for seizure detection. For instance, Gotman [14] proposed the extraction of time-domain features based on the signal amplitude, mean value, and coefficient of variation. Alka et al. [15] suggested the extraction of frequency-domain features centered around the peak frequency and main frequency peak bandwidth. Azami et al. [16] introduced the idea of extracting nonlinear features rooted in multi-scale fuzzy entropy and sample entropy. These extracted features are subsequently fed into various classifiers for seizure detection and classification, such as support vector machines [17], Gaussian mixture models [18], and others. However, manual feature extraction is a labor-intensive process. EEG signals encompass multiple channels, rendering them multi-dimensional, which complicates their processing through these methods. Therefore, the accuracy and sensitivity of traditional machine learning algorithms demonstrate certain limitations when applied to seizure detection and classification tasks.

In recent years, the advancement of deep learning technology has showcased strong generalization and adaptive learning capabilities in the realm of epilepsy research, thereby becoming the leading method in the field of seizure detection and classification. Most studies typically utilize conventional deep learning models, such as recurrent neural networks (RNNs) (Tsiouris et al. and Usman et al. [10,11]) and convolutional neural networks (CNNs) (Saab et al.; Ahmedt-Aristizabal et al. and Dhar et al. [8,9,19]). However, these methods have certain limitations. For example, RNNs neglect the structural features

inherent in brain networks. CNNs presuppose that EEG signals adhere to a Euclidean structure—an assumption that disregards the natural geometry of EEG electrodes and the connectivity within the brain network. While some researchers have endeavored to optimize CNNs using strategies like a neural architecture search or pruning methods (Zhao et al. and Wang et al. [20,21]), guaranteeing model performance remains a significant challenge. As a result, an increasing number of studies have begun employing graph neural networks for epilepsy research. In this context, Shan et al. [22] consider both the adjacency matrix of functional connectivity, derived from multiple EEG channels, and the corresponding dynamics of EEG channel signals simultaneously. They efficiently leverage the constrained spatial topology of functional connectivity and harness discriminative temporal information through 1D convolution.

The aforementioned epilepsy detection methods all presuppose that training and testing data share the same distribution. However, EEG signals across different individuals often exhibit considerable variation. To overcome this obstacle, certain studies have integrated the fusion of transfer learning with deep learning. For instance, Zhang et al. [23] converted the EEG signal into a time–frequency graph and transferred parameters from three existing CNN networks: VGG16, VGG19, and ResNet50. Zhu et al. [24] incorporated meta-learning into transfer learning to reduce the model training time via memory enhancement modules. Some other studies have adopted pre-training models. Tang et al. [4], for example, pre-trained the Encoder of their proposed seizure detection model using an epilepsy sequence prediction task, ultimately achieving superior results in comparison with the transfer learning method.

## 3. Methods

### 3.1. Problem Statement

In this paper, the EEG input data comprise sensor readings from $M$ sensors over $N$ time steps. We use $X$ to denote our input data:

$$X = \{x_1, \cdots, x_i, \cdots, x_N\}, X \in \mathbb{R}^{N \times M} \tag{1}$$

where $x_i \in \mathbb{R}^M$ represents the signal acquired from $M$ sensors at the $i$-th time step, forming an $M$-dimensional vector. For instance, if a segment of an EEG signal contains a 12 s or 60 s interval, the corresponding time steps $N$ would be $12 \times f$ and $60 \times f$, respectively, where $f$ is the sampling frequency.

In this study, we formulate the representation of the EEG signal as a graph denoted by $\mathcal{G} = \{\mathcal{V}, \epsilon, W\}$, where $\mathcal{V}$ corresponds to the set of sensors, $\epsilon$ signifies the set of edges, and $W$ is the weight matrix. It is important to highlight that the weight matrix $W$ is initially unestablished and is intended to be inferred through our novel model. Subsequently, we outline the primary objectives of our investigation as follows:

In the context of seizure detection, our primary objective involves predicting the presence or absence of a seizure within a given EEG clip. Additionally, our focus extends to seizure classification, wherein the aim is to categorize the specific type of seizure present within an EEG clip. Building upon insights from prior research [4], we adopt established methodologies. In order to comprehensively evaluate the performance of our proposed model across different temporal scales, we segment the original EEG signal into clips of 12 s and 60 s duration, corresponding to fast and slow analysis, respectively. This segmentation strategy enables us to explore the model's proficiency in both rapid and prolonged detection as well as classification scenarios.

### 3.2. Framework of the DTGCN Model

Our DTGCN model aims to learn the distribution patterns of epilepsy and uncover the intricate connectivity modes within the brain network. As demonstrated in Figure 2, the proposed DTGCN method contains three key components: the seizure attention module, graph structure learning module, and temporal graph convolution network. Firstly, we

establish the seizure attention module to reconstruct EEG signals, with the goal of emphasizing the distinguishing features between normal and abnormal signals. Subsequently, we integrate the spatial–temporal information within the hidden states of the TGCN and the reconstructions from the seizure attention module to learn the graph structure. Furthermore, the reconstructed signals and the learned weighted matrices serve as input features for the TGCN. Finally, classifiers are implemented to perform the seizure detection and classification tasks. The well-structured framework effectively captures both the temporal dynamics and spatial connectivity patterns within the EEG data.

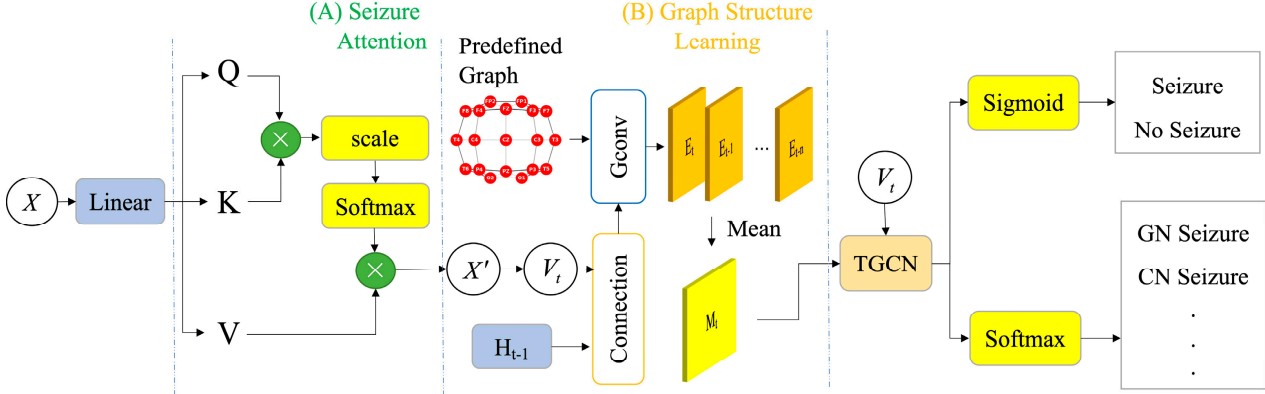

**Figure 2.** Framework of the DTGCN Model. $X$ and $X'$ respectively denote the input and reconstructed output. $V_t$ is a sequence of windows sliced from $X'$. $H_{t-1}$ represents the output of TGCN at the $(t-1)-th$ time step. $E_t$ represents the adjacency matrix of the graph structure generated at the $t-th$ time step. Gconv represents graph convolution. The upper right and lower right two different branches represent seizure detection and classification respectively.

### 3.3. Seizure Attention Module

As shown in Figure 3, in the context of a signal segment featuring epileptic seizures, the coarse-grained label '1' signifies the presence of epileptic seizures within this segment. Meanwhile, the fine-grained label [ 0 0 0 . . . 1 1 1 . . . 0 0 0 ] precisely denotes the specific time points of seizure onset. The previous studies only use the coarse-grained labels in the training process, i.e., the ground truth label of a EEG segment. However, they ignore the fine-grained labels, i.e., the ground truth labels of each time step within one segment [4,5]. The fine-grained labels for each time step reflect the distribution and spread patterns of epilepsy. In order to effectively capture the seizure pattern, we construct a seizure attention module and propose a constrained reconstruction loss based on the fine-grained labels in our training objective function.

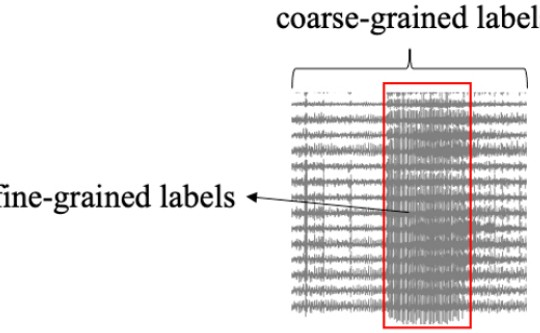

**Figure 3.** Fine-grained and coarse-grained labels.

As shown in Figure 2 (A), the raw EEG signal $X \in R^{N \times M}$ is input into the multi-head attention module, and then we can obtain the reconstructions as follows:

$$\{Q, K, V\} = \{W_Q, W_K, W_V\} * X$$
$$S = Softmax\left(\frac{QK^T}{\sqrt{d_{model}}}\right) \quad (2)$$
$$X^{'} = SV, X^{'} \in \mathbb{R}^{N \times M}$$

where $Q, K, V \in \mathbb{R}^{d_{model} \times N}$ represent the query, key, and value in the self-attention mechanism. $S$ is the attention score and $X^{'}$ represents the reconstructed signal.

To learn the distinguishing features of EEG signals, we add the reconstruction loss and constrained loss based on fine-grained labels between raw signals $X$ and reconstructed signals $X'$. Specifically, we define time steps with high reconstruction errors as anomaly instances, drawing inspiration from the previous research on anomaly detection methodologies [25–27], which can be formulated as $\|x_t - x_t^{'}\|_2 > \tau$, $x_t \in R^M$, and $\tau$ is the optimal threshold. To determine the optimal threshold $\tau$, we employ a dynamic search approach. Within the training dataset, we uniformly sample 50 values spanning from the minimum to the maximum of the $\ell_2$ norm $\|x_t - x_t^{'}\|_2$. From this set of values, we select the threshold $\tau$ that yields the most favorable performance on our validation set. Locate the place with large reconstruction loss as the time step of epileptic seizure. This process can result in a set of fine-grained labels.

Then, we add constraints to the loss to minimize the difference between the true distribution of epilepsy and the model-predicted distribution of epilepsy as follows:

$$T_1 = \left\{t \mid label_{fine-grained} = 1\right\} \quad (3)$$

$$T_2 = \left\{t \mid \|x_t - x_t^{'}\|_2 > \tau\right\} \quad (4)$$

$$L_{re} = \|X - X^{'}\|_2 + \left(crad(T_1) - crad(T_2)\right)/n \quad (5)$$

where $T_1$ denotes the aggregation encompassing all the time steps corresponding to the actual seizures, $T_2$ denotes the ensemble encompassing all the temporal steps attributed to the projected seizures, $label_{fine-grained}$ represents fine-grained labels, and $crad$ represents the number of elements in the collection.

The first term of $L_{re}$ is the standard reconstruction loss, which preserves the informative features inherent in the raw EEG signals. The second term is the regularization, which can highlight the discrepancy between seizure signals and normal signals and thus generate discriminative features for further modules. The architecture of the seizure attention model empowers the framework with robust capabilities for discriminating between normal and abnormal data.

### 3.4. Graph Structure Learning Module

The intricate connectivity of neurons within the brain is inherently influenced by diverse spatiotemporal relationships. Therefore, it is natural to describe the connection network of neurons in the brain from a dynamic perspective. In response, we designed a network that generates dynamic graphs for constantly updating dynamic adjacency matrix $E_t$. This dynamic adjacency matrix accounts for the fluctuations in the network's connectivity over time. In order to effectuate this, we transform the original time series $X^{'}$ into a sequence of windows $V_t$. For a window of length $K$, the input after adding the window is expressed as:

$$V_t = \left\{x_{t-K+1}^{'}, \cdots, x_{t-1}^{'}, x_t^{'}\right\} \quad (6)$$

$$I_t = V_t \parallel H_{t-1} \quad (7)$$

where $\|$ represents the concatenation operation, $H_{t-1}$ represents the hidden-state output by the gated recurrent unit at the previous time step, and we regard $I_t$ as a dynamic node feature input to the graph convolution module:

$$DF_t = \theta_G\{I_t\} \tag{8}$$

where $\theta_G$ represents the graph convolution on a predefined graph G, which can carry out the message-passing process of the dynamic node features. We choose Dist-Graph [4] as our predefined graph, and its adjacency matrix is calculated from the distance between nodes.

In practice, we generate the dynamic features of the source node $DE_t^1$ and the destination node $DE_t^2$, respectively, and then calculate the dynamic adjacency matrix $E_t$ according to the similarity between the nodes. While considering that, the change rate of the brain network should not be too fast. For time step $t-1$ and time step $t$, our graph structure should not change much. So, we average the dynamic adjacency matrix of $N$ consecutive time steps to obtain the adjacency matrix $M^t$ of the node graph structure at time t, denoted as follows:

$$DE_t^1 = tanh(\alpha DF_t^1) \tag{9}$$

$$DE_t^2 = tanh(\alpha DF_t^2) \tag{10}$$

$$E_t = RELU\left(tanh(\alpha(DE_t^1 DE_t^2 - DE_t^2 DE_t^1))\right) \tag{11}$$

$$M^t = (E_{t-n} + \cdots + E_{t-1} + E_t)/n \tag{12}$$

where $\alpha$ is a hyper-parameter to control the saturation rate of the activation function to ensure the sparsity of the graph.

*3.5. Temporal Graph Convolution Network*

We adopt the TGCN [13] to model the spatiotemporal dependence of EEG signals, which is a neural network that combines the graph convolution and gated recurrent unit (GRU), originally developed for traffic prediction in urban road networks. Specifically, the graph convolution part is used to learn complex topological structures for learning spatial correlation, and the gated recurrent unit is used to learn dynamic changes in EEG signals for temporal correlation. In terms of the specific implementation, the convolutional section of the GRU is modified as the graph convolutional section:

$$u_t = \sigma\left(W_u\left[\theta_{G'}(I_t), h_{t-1}\right] + b_u\right) \tag{13}$$

$$r_t = \sigma\left(W_r\left[\theta_{G'}(I_t), h_{t-1}\right] + b_r\right) \tag{14}$$

$$c_t = tanh\left(W_c\left[\theta_{G'}(I_t), r_t * h_{t-1}\right] + b_c\right) \tag{15}$$

$$h_t = u_t * h_{t-1} + (1 - u_t) * c_t \tag{16}$$

Among them, $u_t$ and $r_t$ are the output of the update gate and the reset gate at time $t$; $w$ and $b$ represent the weight and bais trainable parameters; $h_t$ is the output of the TGCN module at time $t$ and is sent to the graph generation module and TGCN-cell as the hidden state at time $t + 1$. We take the output at the last moment as the final output vector of the model and send it to the downstream classification. We employ cross-entropy to compute the loss for the epilepsy detection and classification tasks:

$$L(\theta) = -log \ p\left(F(X) = e|\theta_m\right) + \lambda L_{re} \tag{17}$$

where $F(X)$ represents the classification result of the EEG fragment $X$ output from the DTGCN model, and $e$ is the real label of $X$, which represents different meanings in seizure detection and classification: the seizure detection task represents whether there is epilepsy, and the seizure classification task represents the type of epilepsy. $\theta_m$ represents the model parameters that need to be trained, $L_{re}$ represents the regularized reconstruction loss

calculated by the seizure attention network, and $\lambda$ represents the weight factor of this loss item.

## 4. Results Analysis

### 4.1. Experimental Dataset

We used the public Temple University Hospital EEG Seizure Corpus (TUSZ) v1.5.2 [6,7], which is the largest public EEG seizure database to date with 5612 EEGs, 3050 annotated seizures from clinical recordings, and eight seizure types. Following the settings of Tang et al. [4] in the TUSZ epilepsy detection experiment: we include 19 EEG channels in the standard 10–20 system, set the sampling frequency of the EEG signal to 200 Hz. In the process, we reclassified epilepsy categories within the original dataset into four distinct classes: CF (comprising focal non-specific (FN), simple partial seizure (SP), and complex partial seizure (CP)), GN (representing generalized non-specific seizure), AB (indicating absence seizure), and CT (denoting tonic-clonic seizure with tonic seizure combined into a single CT seizure class). To evaluate model generalizability to unseen patients, we exclude five patients from the official TUSZ test set who exist in both the official TUSZ train and test sets. The dataset statistics are shown in Table 1. Column EEG Files indicates the number of EEG files, and the percentage of files containing epilepsy in the total files is indicated in parentheses. Column Patients indicates the number of patients tested, and the brackets indicate the percentage of patients with epilepsy. The brackets in the total duration column indicate the proportion of epilepsy duration to the total EEG duration. The brackets in the CF seizure column indicate the number of patients with CF seizure.

**Table 1.** Summary of data in train and test sets of TUSZ v1.5.2. used in our study.

| | EEG Files (% Seizure) | Patients (% Seizure) | Total Duration (% Seizure) | CF Seizures (% Seizure) | GN Seizures (Patients) | AB Seizures (Patients) | CT Seizures (Patients) |
|---|---|---|---|---|---|---|---|
| Train Set | 4599 (18.9%) | 592 (34.1%) | 45,174.72 min (6.3%) | 1868 (148) | 409 (68) | 50 (7) | 48 (11) |
| Test Set | 900 (25.6%) | 45 (77.8%) | 9031.58 min (9.8%) | 297 (24) | 114 (11) | 49 (5) | 61 (4) |

### 4.2. Experimental Setup

We divided the train set from the official TUSZ dataset randomly into a train set and a verification set, maintaining a ratio of 9:1. Our experiments were conducted on a single RTX 3080 graphics card, and we utilized the Adam optimizer within the PyTorch framework. The learning rate was set at $5 \times 10^{-5}$, and we trained for 100 epochs. To ensure graph structure sparsity, we retained only five edges for each node. Should the verification loss fail to decrease consistently for five consecutive epochs, the model's training was terminated prematurely.

According to some recent studies [4,5,28], we used the AUCROC and Weighted F1-score as the main evaluation metrics for seizure detection and classification, respectively. In order to comprehensively compare our DTGCN model with general CNN/RNN approaches and other graph theory-based methods, we chose the following baselines:

(1) LSTM [10]: A variant of an RNN with gating mechanisms.
(2) Dense-CNN [9]: A previous state-of-the-art CNN for seizure detection.
(3) CNN-LSTM [8]: A CNN and RNN framework enhanced by using external memory modules with trainable neural plasticity.
(4) STS-HGCN-AL [28]: A model that extracts hierarchical graphs via a spectral–temporal convolutional neural network and variant self-gating mechanism and then through the hierarchical graph network to capture the spatiotemporal characteristics of the rhythm.
(5) Dist-DCRNN and Corr-DCRNN with pre-training [4]: They are two variants of the DCRNN [29], which, respectively, adopt two different mapping methods: mapping based on the real distance between electrodes and the correlation between EEG signals from different electrodes.

(6)   PLV+GCNN and Spatial+GCNN [12]: They represent models that integrate temporal features extracted from individual EEG signals with short- and long-range spatial interdependencies among EEG channels. PLV and Spatial are two variants of graph structures.

*4.3. Overall Performance*

Table 2 presents overall experimental results. DTGCN outperforms all the baseline models expect Dist-DCRNN on 12-s seizure classification. Our model achieves a slightly higher score compared to Dist-DCRNN and Corr-DCRNN, and significantly outperforms other methods. It's important to note that while LSTM can capture sequence information and CNN can capture certain spatial structures, they do not accurately represent the complex structure of the human brain. STS-HGCN-AL, Dist-DCRNN, and Corr-DCRNN encode structural information within a graph framework. However, research has demonstrated that during a seizure event, the connectivity among a substantial number of brain neurons undergoes alterations [30]. While GCNN does consider long-range correlation, it also captures spatial correlation through static graph representations. These models tend to overlook fine-grained label information and the dynamic nature of brain networks. Even though our model's performance on 12-second EEG signals after segmentation doesn't match that of Dist-DCRNN, it's worth noting that Dist-DCRNN benefits from pre-training via self-supervised tasks. In contrast, our DTGCN is trained from scratch, and its training time is notably shorter than that of Dist-DCRNN.

**Table 2.** Seizure detection and seizure classification results.

| Model | Seizure Detection AUCROC | | Seizure Classification Weighted F1-Score | |
|---|---|---|---|---|
| | **12 s** | **60 s** | **12 s** | **60 s** |
| LSTM | 0.629 | 0.586 | 0.576 | 0.601 |
| Dense-CNN | 0.786 | 0.715 | 0.652 | 0.679 |
| CNN-LSTM | 0.749 | 0.682 | 0.641 | 0.666 |
| STS-HGCN-AL | 0.809 | 0.772 | 0.707 | 0.714 |
| Corr-DCRNN | 0.856 | 0.843 | 0.723 | 0.741 |
| Dist-DCRNN | 0.861 | 0.875 | 0.747 | 0.750 |
| PLV + GCNN | 0.867 | 0.871 | 0.728 | 0.734 |
| Spatial + GCNN | 0.852 | 0.850 | 0.724 | 0.727 |
| DTGCN(ours) | 0.873 | 0.889 | 0.742 | 0.759 |

As depicted in Figure 4, our evaluation of the effectiveness of each model in classifying uncommon forms of epilepsy involves the presentation of confusion matrices. These matrices encapsulate the outcomes of both the baseline models and our DTGCN model within the context of 60 s seizure classification. In an effort to provide an unbiased perspective, each row within the confusion matrices has undergone normalization by division with the total number of examples corresponding to the respective class. In the graphical representation, the horizontal axis delineates the epilepsy category predicted by the model, while the vertical axis signifies the authentic epilepsy category label. Notably, our DTGCN model demonstrates a marked advancement of 4 points in contrast to the finest baseline in the AB category of rare epilepsy. Similarly, a noteworthy improvement of 5 points is evident in the CT category of rare epilepsy in comparison with the leading baseline. And the comprehensive classification performance has also improved.

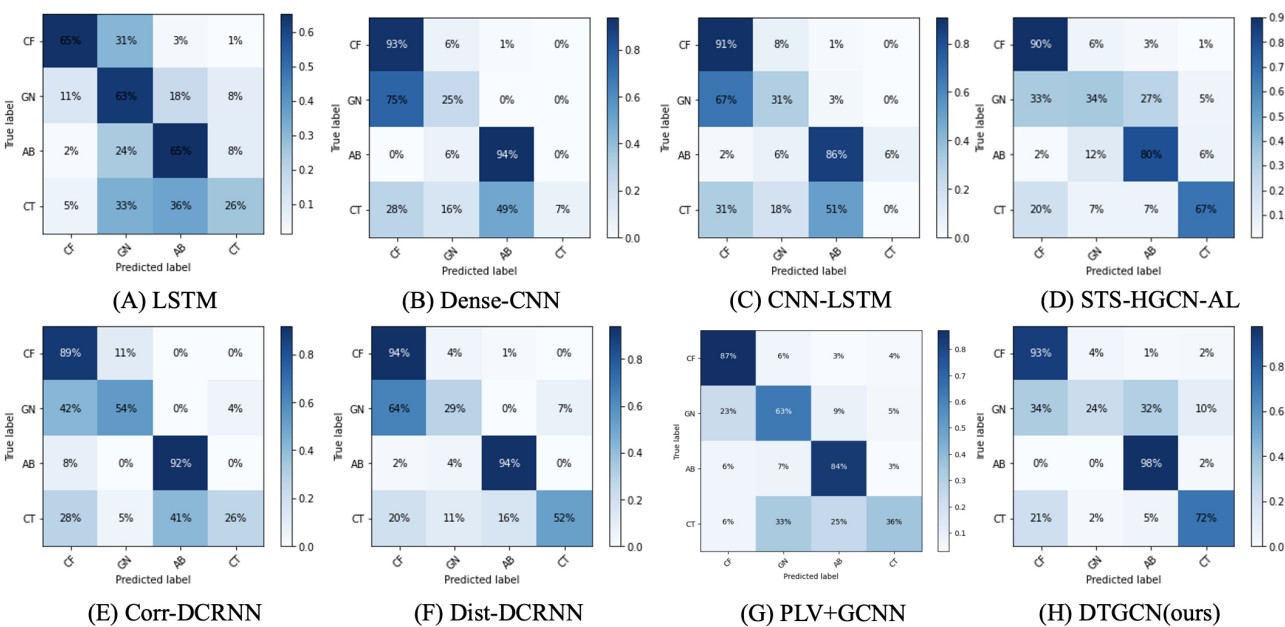

**Figure 4.** Confusion matrices of the baselines and DTGCN for 60 s seizure classification. The intensity of the color increases with the improvement of prediction accuracy, with darker hues corresponding to higher levels of accuracy.

### 4.4. Ablation Study

Since the Seizure Attention module and the Graph Structure Learning module are important components of our DTGCN, we study the impact of these two parts on the model performance. Table 3 shows the performance of DTGCN after we remove the Seizure Attention module and Graph Struct Learning module in DTGCN respectively. DTGCN is our original model, DTGCN w/o sz-atten removes the Seizure Attention module of the proposed framework. TGCN w Dist-graph removes the Graph Struct Learning module and the pre-defined graph is Dist-Graph. The experimental results presented are the averages of three independent experiments, and we provide the corresponding standard deviations.

**Table 3.** Comparison of ablation experiments.

| Model | Seizure Detection AUC/STD | | Seizure Classification Weighted F1/STD | |
|---|---|---|---|---|
| | **12-s** | **60-s** | **12-s** | **60-s** |
| DTGCN | 0.873/12.45 | 0.889/11.41 | 0.742/15.59 | 0.759/12.48 |
| DTGCN w/o sz-atten | 0.869/10.85 | 0.883/8.23 | 0.726/12.17 | 0.732/9.93 |
| TGCN w Dist-graph | 0.861/9.94 | 0.856/9.20 | 0.737/9.72 | 0.729/8.34 |

Upon the exclusion of the Seizure Attention module, we observed a modest decline in the performance of the epilepsy detection task. In contrast, the epilepsy classification task exhibited a notable reduction in performance. This discrepancy can likely be attributed to the elevated significance of the epilepsy distribution information inherent in the fine-grained labels, particularly in the context of the classification task. This observation emphasizes the intrinsic value of our fine-grained labeling approach, as it effectively encapsulates pertinent details related to epileptiform patterns. Upon the omission of the Graph Structure Learning module, we opted to directly feed the pre-defined graph structure, denoted as Dist-Graph, into the TGCN module. This decision led to a pronounced plummet in the model's performance. This notable decline serves as an indication that the dynamic graph, in contrast to the static graph, aligns more coherently with the inherent structural information present within the EEG signal. In summary, the absence of either the Seizure Attention module or the Graph Struct Learning module distinctly influences

the achieved outcomes. This interdependence underscores the efficacy of our proposed methodology in enhancing the overall performance.

### 4.5. Effects of Parameters

In this section, we study the effects that different parameters and factors that can have impact on the performance of DTGCN. These experimental results are the average results of three experiments.

The first factor we study is how DTGCN responds to different data window sizes K. The window size has an impact on the types of seizures that can be detected, and directly affects the speed of epilepsy detection and classification, because larger windows mean more computation. Figure 5A summarizes the obtained results using five different window sizes: K ∈ [1, 5, 10, 20, 50]. The results show that DTGCN's performance is relatively insensitive to window size and has a constant performance. It may be that our seizure attention module has already extracted the global latent information of this segment of the EEG signal, so the additional information captured is limited when faced with different window sizes. For our experiments, we selected a window size of 10. This is the best trade-off between the consumption of training resources and model performance.

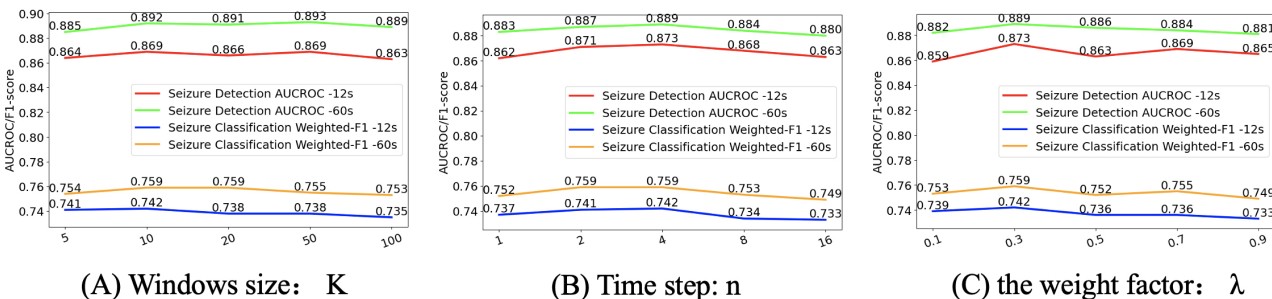

(A) Windows size： K        (B) Time step: n        (C) the weight factor： λ

**Figure 5.** The impact of parameter changes on model performance: (**A**) the window size K, (**B**) the time step n that needs to be averaged, and (**C**) the weight factor $\lambda$ of the KL divergence loss item.

The second aspect we examine pertains to how DTGCN adapts when averaging over different time steps, denoted as 'n'. As described in the previous section, considering that the rate of change of the brain network should not be too fast, we average the elements of the dynamic adjacency matrix for n consecutive time steps to obtain the adjacency matrix of the graph structure at time *t*. Figure 5B presents the obtained results for five different values of n ∈ [1, 2, 4, 8, 16]. The best result was achieved for n = 4. When n is small, the structure of the brain network graph changes too quickly, and the structural information cannot be effectively fused with the temporal information, resulting in poor performance. In another extreme case, if we take a larger n, the effect is similar to that of a static graph, and the graph structure does not change much, resulting in a decrease in performance. At the same time, from the experimental results, the mid-range value of n seems to have little effect on the performance, showing relatively high and stable AUCROC and Weighted F1-Scores.

Finally, we investigate the effect of the weight factor $\lambda$ of the reconstruction loss term, with a larger $\lambda$ corresponding to greater emphasis on the seizure attention module in the model. Figure 5C shows the results of the different $\lambda \in$ [0.1, 0.3, 0.5, 0.7, 0.9]. When $\lambda$ is 0.3, the model achieves the best results.

### 4.6. Visualization Results and Interpretability

To investigate the practical significance of the learned graph structure and to compare the effectiveness of our method, we visually analyzed the average adjacency matrix produced by our DTGCN model alongside the best-performing DCRNN model from the baseline across the entire test set. This visualization encompasses instances of both correctly and incorrectly detected instances.

As shown in Figure 6A–C represent average connectivity matrix for no seizure, generalized seizure and focal seizures. Figure 6D represents the difference between focal seizure

and non-seizure adjacency matrices and Figure 6E represents the difference between generalized seizure and non-seizure adjacency matrices. Comparing (D,E) of DTGCN in Figure 6, it can be found that the difference between the adjacency matrix of generalized seizure and no seizure is greater than that of focal seizure and no seizure, which indicates that generalized seizure can disrupt normal brain connections to a greater extent than focal epilepsy. This is consistent with the previous description in the literature: brain functional connectivity is significantly altered in global seizures [31]. Comparing (I,J) of DCRNN in Figure 6, we find that there is no obvious difference in their graph structures. In summary, The graph structure learned by our model can reflect epilepsy categories to some extent.

DTGCN (ours)

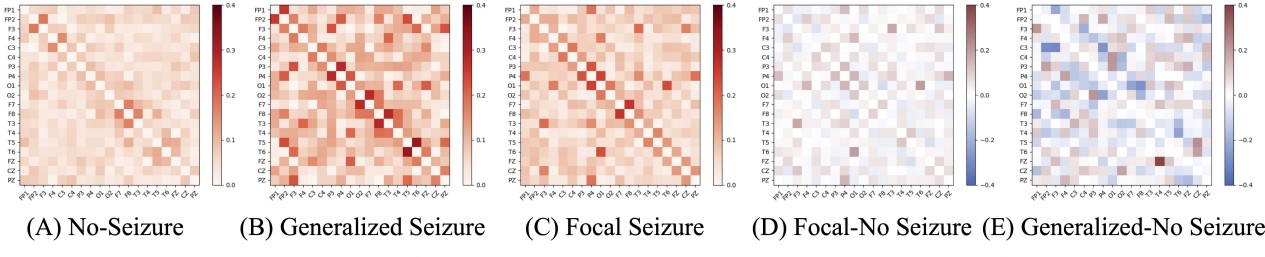

(A) No-Seizure　　(B) Generalized Seizure　　(C) Focal Seizure　　(D) Focal-No Seizure　(E) Generalized-No Seizure

DCRNN

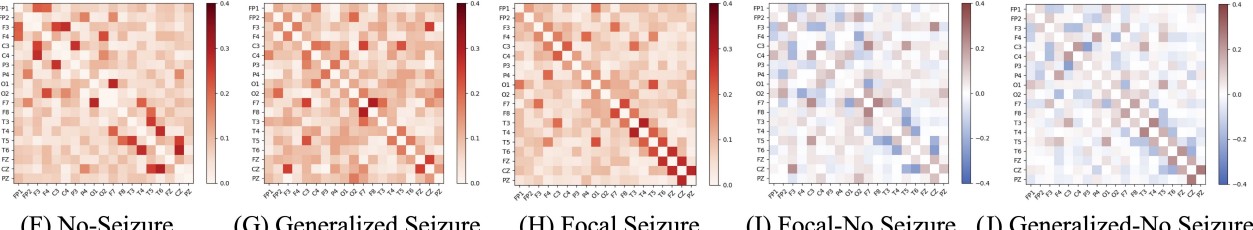

(F) No-Seizure　　(G) Generalized Seizure　　(H) Focal Seizure　　(I) Focal-No Seizure　(J) Generalized-No Seizure

**Figure 6.** Mean adjacency matrix for EEG in the test set for (**A**,**F**) non-seizure EEG clips, (**B**,**G**) generalized seizures, (**C**,**H**) focal seizures, (**D**,**I**) difference between focal seizure and non-seizure adjacency matrices, and (**E**,**J**) difference between generalized seizure and non-seizure adjacency matrices.

## 5. Conclusions

In this study, we propose a novel learning model DTGCN designed for the detection and classification of seizures through the fusion of dynamic graphs and gated recurrent units. Additionally, we present a method that leverages fine-grained label information to effectively model the distribution and diffusion patterns inherent in epilepsy. Notably, our model circumvents the need for a pre-training phase, substantially improving the performance of automated seizure detection and classification. Of particular significance, our model acquires a coherent graph structure that carries meaningful implications by accurately capturing the distinctions among various seizure categories. This feature contributes to the overall interpretability of our model. Our method eliminates the pretraining step and effectively improves the performance of automated seizure detection and classification. We anticipate extending our model to encompass other multi-channel sequential tasks, capitalizing on domain-specific knowledge to refine our graph structure learning module.

**Author Contributions:** Methodology, G.W. and H.Z.; writing—review and editing, G.W., K.Y., X.W. and S.S. All authors have read and agreed to the published version of the manuscript.

**Funding:** This work is partially supported by the National Natural Science Foundation of China under the Grant No. 62371057 and 61601046, the 111 Project of China under the Grant No. B08004, and the BUPT Excellent Ph.D. Students Foundation under the Grant No. CX2022149.

**Informed Consent Statement:** Informed consent was obtained from all subjects involved in the study.

**Data Availability Statement:** We used the public Temple University Hospital EEG Seizure Corpus (TUSZ) v1.5.2.

**Conflicts of Interest:** The authors declare no conflicts of interest.

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
