# Peer review of "Time-Series Anomaly Detection Based on Dynamic Temporal Graph Convolutional Network for Epilepsy Diagnosis"

_bioengineering, doi:10.3390/bioengineering11010053_

Round 1

Reviewer 1 Report

Comments and Suggestions for Authors

In this manuscript, the Authors propose and demonstrate a new model for epilepsy diagnosis and epilepsy stage classification that uses a dynamic temporal graph convolutional network approach. Interpreting electroencephalography (EEG) signals for diagnostic purposes can be difficult and time consuming, thus it is necessary to develop precise and automated models to facilitate the process. Traditional methods often neglect the coherent information between EEG signal segments and instead focus only on specific EEG signal periods for diagnosis. The model described by Wu et al. incorporates temporal dynamics and spatial connectivity patterns from EEG by including: (1) Seizure Attention Module, which distinguish “normal” and “abnormal” data input; (2) a Graph Structure Learning Module, which updates the network’s connectivity in a dynamic fashion; and (3) a Temporal Graph Convolution Network, which models the spatiotemporal dependence of EEG signals. The Authors demonstrate from their test performance outcomes that all three modules are necessary to effectively detect and classify seizure.

While the model and corresponding test results are well-presented, the arguments for research novelty and significance need to be improved. I am concerned especially with the particular literature references made in the present manuscript as they seem to cherry-pick only previous works that used more traditional methods. As the Authors have eluded, without appropriately referencing, there are many recent works that have also described models which utilize dynamical graph convolutional neural networks and spatial–temporal graph convolutional network approaches, such as DOI: 10.1109/TAFFC.2018.2817622, DOI:10.1002/hbm.25994, among many others. There are even many publications that describe models using graph convolutional neural network for the detection of epilepsy, such as DOI: 10.1016/j.cmpb.2022.106950. Furthermore, although I would not stress this point under most circumstances, this manuscript requires major writing and grammatical revisions, especially within the first two sections of the manuscript.

However, technical improvements aside, the manuscript is solid and novel methods for seizure detection are urgently needed to improve epilepsy diagnosis and relapse prediction. Therefore, I recommend the manuscript be accepted only after the following major concerns are addressed:

1. The title of the manuscript contains a typo: “temproal".

2. The Authors need to compare their work with more closely related publications and state what are the unique strengths and weakness of their described model compared with the existing precedent.

3. There are many typos and grammatical errors that need to be corrected in the manuscript. Besides the main title, there is the title of Section 3.5, "Nework". Also, Figure 2 has the exact same caption as Figure 1. Moreover, there are a few places that lack proper citations, such as line 45-50, 123, 172, and many other places. Please address these concerns.

4. There appears to be a discrepancy in the Author list between the online submission portal and the actual manuscript. Please doublecheck.

Comments on the Quality of English Language

Please refer to my comments above, regarding the manuscript. To reiterate: there are many typos and grammatical errors that need to be corrected in the manuscript. Besides the main title containing a typo, there is the title of Section 3.5, "Nework". Also, Figure 2 has the exact same caption as Figure 1. Moreover, there are a few places that lack proper citations, such as line 45-50, 123, 172, and many other places. Please address these concerns.

Reviewer 2 Report

Comments and Suggestions for Authors

What are the influences of the different focal seizure locations on your training data set? 
Please explain the acronyms of CF, GN, AB and CT seizures. 
